# A Study of Cross-Linguistic Speech Emotion Recognition Based on 2D Feature Spaces

**Gintautas Tamulevičius [1], Gražina Korvel [1] , Anil Bora Yayak [2] , Povilas Treigys [1], Jolita Bernatavičienė [1],\* and Bożena Kostek [3]**

[1]  Institute of Data Science and Digital Technologies, Vilnius University, 01513 Vilnius, Lithuania; gintautas.tamulevicius@mif.vu.lt (G.T.); grazina.korvel@mif.vu.lt (G.K.); povilas.treigys@mif.vu.lt (P.T.)

[2]  Department of Electrical and Electronics Engineering, Izmir Katip Celebi University, İzmir 35620, Turkey; abora.yayak@gmail.com

[3]  Faculty of Electronics, Telecommunications and Informatics, Gdansk University of Technologies, 80233 Gdańsk, Poland; bokostek@audioakustyka.org

\*  Correspondence: jolita.bernataviciene@mif.vu.lt; Tel.: +370-5-210-9315

**Abstract:** In this research, a study of cross-linguistic speech emotion recognition is performed. For this purpose, emotional data of different languages (English, Lithuanian, German, Spanish, Serbian, and Polish) are collected, resulting in a cross-linguistic speech emotion dataset with the size of more than 10.000 emotional utterances. Despite the bi-modal character of the databases gathered, our focus is on the acoustic representation only. The assumption is that the speech audio signal carries sufficient emotional information to detect and retrieve it. Several two-dimensional acoustic feature spaces, such as cochleagrams, spectrograms, mel-cepstrograms, and fractal dimension-based space, are employed as the representations of speech emotional features. A convolutional neural network (CNN) is used as a classifier. The results show the superiority of cochleagrams over other feature spaces utilized. In the CNN-based speaker-independent cross-linguistic speech emotion recognition (SER) experiment, the accuracy of over 90% is achieved, which is close to the monolingual case of SER.

**Keywords:** speech analysis; speech emotion recognition; 2D feature spaces

## 1. Introduction

The discovery of expressivity discovery in speech is one of the unresolved challenges. Among the relevant topics belonging to the expressivity are the issues related to the recognition of emotion. The recognition of emotion is a part of sentiment analysis [1,2]. Emotions are expressed both through verbal and non-verbal processes. The information derived from the speech signal, facial expressions, gestures, eye contact, and physiological status (e.g., EEG, EMG, respiration signals) is considered in studies in the area of recognition of emotions [3–5]. Emotion recognition based on the human speech, however, has the most applications, not only the evident ones related to the direct communication between people or human–computer interaction but also concerning the affective state recognition, e.g., recognizing social cognitive skills, diagnosing emotion deficits, psychotherapy, etc.

The speech emotion recognition task is a classification task by nature: the unknown emotional speech pattern is assigned to a particular emotional class based on the expressive attributes (called features) in the speech signal. Therefore, the first stage of the recognition process is to search for and extract such features. In the next step, the extracted features are classified to identify the emotional class that the analyzed speech utterance belongs to. Most current studies report speech emotion classification rates of 70%–90% [6].

The results depend heavily on the analyzed language, the number of emotions, the speaker's mode, and the extracted features. A few main research directions could be distinguished in speech emotion recognition over the last years. The research of the speech emotion recognition was initiated with the intensive exploration of emotion features within the audio signal: various combinations of temporal, spectral, cepstral, energy features describing the rate, intensity, and prosody of the emotional speech. Furthermore, these feature sets were extended with various statistical derivatives of the primary features. This resulted in vast sets of a few thousand of features leading to the inevitable need to reduce the feature dimensions. For this purpose, numerous feature selection and transformation techniques were proposed.

The rise of the deep-learning paradigm enabled researchers to analyze the speech signal directly, omitting the signal analysis stage. The multimodal emotion recognition was implemented by combining a speech signal with additional emotional attribute sources such as facial expression, gesture, body pose, electrocardiography, electroencephalography signals. However, the analysis of the speech signal remains one of the most relevant approaches to recognize the speakers' emotion.

Global services, containing a cross-linguistic environment, bring new challenges in the speech emotion recognition (SER) tasks. Different cultural contexts, divergent acoustic properties, and their variation within languages, the lack of multilingual or cross-language datasets make the task of cross-linguistic SER non-trivial, and the progress in this field is below our expectations [7].

Cross-linguistic emotion recognition is challenging for several reasons: it is unclear which features of speech have the highest potential in distinguishing emotions. Moreover, different language, speakers, speaking in different styles, and speaking rates affecting the most commonly extracted speech features, such as pitch and energy contours, are important for emotion analysis [8]. Moreover, the datasets of recordings of the actors' statements or spontaneous speech can be a source of information. Still, it is rather spontaneous speech that should be analyzed in the context of emotions as the professional stage performers tend to exaggerate some features of the voice to make emotions more distinguishable.

There is also the question of whether non-native speakers transfer the same emotion expression pattern from their mother tongue to another language [9]. Within this context, a well-known problem, such as the so-called foreign language anxiety, i.e., how to express emotions in a foreign language, occurs [9].

The majority of researchers in SER have focused on monolingual [10–12] or multiple emotion classification, while the cross-linguistic speech emotion recognition is much less frequented [13–15].

The classification performance of conventional (state-of-the-art) as well as the deep-learning classifiers, employed for the task of speech emotion recognition on Lithuanian [16], English [17–20], Serbian [21], Spanish [22], German [23], Polish [24,25], Greek [26], Chinese [20,27], French [28], and Italian [29,30] datasets are presented in Table 1.

According to the analyzed literature, there are two ways of classifying emotions:

- Methods based on a two-pass classification scheme consisting of language identification and language-specific speech emotion recognition [14,41];
- Direct cross-linguistic emotion recognition using different language databases, analyzing speech signal and recognizing emotions based on these features or parameter description that results from redundancy checking (e.g., PCA, principal component analysis-based) [13,42,43].

The speech features for emotion recognition can be grouped into several categories: local, global, continuous features, qualitative features, spectral features, and TEO (Teager energy operator)-based on features [8,10]. In our research, we have used four different 2D feature spaces, which are as follows: cochleagrams, spectrograms, mel-cepstrograms, and fractal dimension-based. A comparison of the results obtained in the conducted research with the results available in the literature is also made.

The recognition of emotion is generally performed using feature vectors extraction and the classical machine learning algorithms, including random forest, the support vector machines, k-nearest neighbors, various types of neural networks, hidden Markov models, and their variations [33].

In recent years, the deep-learning algorithms brought new opportunities to machine learning. A very promising property of deep neural networks (DNNs) is that they can learn high-level invariant features from raw data, which is potentially helpful for emotion recognition [13,14,44–46]. Most of the emotion classification experiences come out, however, from the speech recognition tasks. For example, convolutional DNN delivers good results in the recognition in isolated Lithuanian words [47]. Therefore, we apply the same strategy in the current experiment while investigating archetypical primary emotions: anger, happiness, fear, sadness, and neutral [48]. These emotions are the most apparent and distinct in human life.

The paper is organized as follows: it starts with the data preparation and the details on the analysis of the methodology presentation, then the experiment results are presented and discussed. The concluding remarks include the future development of this research.

**Table 1.** Examples of classification performance for the task of speech emotion recognition on datasets analyzed (CNN—convolutional neural network; *k*NN—*k*-nearest neighbor classifier; LSTM—long short-term memory network; SVM—support vector machines, RF—random forest classifier; RNN—recurrent neural network; GMM—Gaussian mixture model).

| Dataset and the Number of Emotions | Classifier | Task | Refs | Accuracies (%) |
|---|---|---|---|---|
| Lithuanian (5) | *k*NN | Monolingual | Liogienė and G. Tamulevičius [31] | 81.7 |
| English SAVEE (7) | Majority voting between multi-class SVM, RF, and Adaboost | Monolingual | Noroozi et al. [32] | 75.71 |
| English RAVDESS (7) | SVM | Monolingual | Bhavan et al. [33] | 75.69 |
| Serbian (5) | 3DEC hierarchical classifier | Monolingual | Hassan and Damper [34] | 94.7 |
| Spanish (7) | RNN | Monolingual | Kerkeni et al. [35] | 91.16 |
| German (4) | GMM + SVM | Monolingual | Vlasenko et al. [36] | 89.90 |
| | Adaptive neuro-fuzzy inference systems (ANFIS) | Monolingual | Li and Akagi [13] | 93.00 |
| | Adaptive neuro-fuzzy inference systems (ANFIS) | Multilingual | Li and Akagi [13] | 91.00 |
| Polish (6) | Majority voting between multi-class SVM, RF, and Adaboost | Monolingual | Noroozi et al. [32] | 88.33 |
| Greek AESDD (5) | SVM | Monolingual | Vryzas et al. [37] | 60.8 (max acc.) |
| | CNN | Monolingual | Vryzas et al. [3] | 69.2 (max acc.) |
| Chinese IEMOCAP (5) | CNN + LSTM | Monolingual | Etienne et al. [38] | 65.3 |
| Italian EMOVO (7) German (4) | GMM | Cross-language | Ntalampiras [15] | 70.1 |
| German, Chinese, and Italian (5) | CNN and bi-directional LSTM with an attention | Multilingual | Fu et al. [39] | 61.14 (for Chinese) 69.26 (for German) 34.50 (for Italian) |
| English and French (2) | Attentive CNN | cross-lingual | Neuman and Vu [40] | From 47.5 to 61.3 |
| English and French (2) | Attentive CNN | multilingual | Neuman and Vu [40] | From 49.3 to 70.1 |

## 2. Methodology

In this study, we investigated the speech emotion classification based on deep learning. The classification is performed by providing the convolutional neural network (CNN) algorithm with two-dimensional feature maps extracted from emotional speech utterances.

### 2.1. Description of Feature Spaces

Although neural networks can operate with one-dimensional data, our experimental results have shown that a higher discriminative power can be obtained for the 2D feature spaces [48,49]. The feature maps are presented as matrices of short-time features converted into greyscale images. All the images obtained by applying specific feature space conversion were scaled to the same size. The following speech signal feature spaces were chosen to be investigated: spectrograms, mel-cepstrograms, cochleagrams, and fractal dimension-based features.

The conventional scheme of speech signal processing and feature extraction was implemented: the speech signal is segmented and windowed (Hamming window was applied), the feature vector is extracted for each segment, thus obtaining the sequence of feature vectors. Arrangement of these vectors into arrays gives us two-dimensional feature maps that can be converted to bitmap images. The images are the outcome of the speech signal analysis and are delivered into CNN to make a decision on the encoded emotion.

The analysis parameters were as follows: the length of the analysis frame was 512 samples; an overlap of 400 samples was applied with the truncation of the last frame.

### 2.1.1. Spectrograms

A spectrogram is constructed from a series of the short-time spectrum of the signal frames. For this purpose, a discrete Fourier transform (DFT) was applied. This transform is given by the following formula:

$$X_k(m) \; = \; \sum_{n=0}^{N-1} x_k(n)\omega(n)e^{(-2\pi j)\frac{nm}{N}} \tag{1}$$

where $X_k(m)$ ($m = 0, \dots, M-1$) are DFT coefficients of frame $k$, $M$ is the number of DFT coefficients, $x_k(n)$ are samples of the analyzed phoneme $k$-th short-time segment, $N$ is the number of samples in the frame, $w(n)$ is the window function.

In order to obtain a spectrogram with all values equally significant, a logarithm-based dynamic range compression of the magnitude spectrum was applied:

$$Sk\,(m) \; = \; log\big|Xk\,(m)\big| \tag{2}$$

where $Xk\,(m)$ ($m = 1, \dots, M$) are Fourier transform coefficients of frame $k$, and $M$ is the number of these coefficients.

### 2.1.2. Mel-Cepstrograms

In the spectrogram, linear frequency scaling is used. Meanwhile, the mel-frequency scale is a quasi-logarithmic scale that approximates the resolution of the human auditory system. The cepstrogram is created according to the classical calculation scheme: DFT-based spectrum is processed by a filter bank (modeling the human auditory system), then a logarithm-based transformation is applied with a subsequent cosine transform. In this research, we applied 13 triangular filters, thus obtaining a 13th-order mel-cepstrum. Cepstral vectors are organized into columns, thus obtaining 2D mel-cepstrograms, i.e., cepstral feature maps.

### 2.1.3. Cochleagrams

Lyon claims that the spectrogram image has too few dimensions to be a full auditory image [50]. In contrast, cochleagram has a unique orthogonal axis to the frequency axis, which gives us this image. In the literature, a cochleagram is a speech representation that shows how our brain processes information received from the ear [50,51]. It is also treated as a computational model of the peripheral auditory system [52].

The cochleagram is constructed by performing auditory filtering. In this study, a bank of gammatone filters covering the range of 20 Hz to 20 kHz is used.

The impulse response of the gammatone filter can be expressed by the following formula [53]:

$$g(t) = At^{n-1}e^{-2\pi Bt}\cos(2\pi f_0 t + \varphi) \tag{3}$$

where $n$ is the filter order (we set $n = 4$), $\varphi$ is the phase (we use phase equal to minimum phase), $B$—the filter bandwidth, $f_0$—the center frequency, $t$—the time, and $A$ refers to the filter amplitude.

The filter amplitude $A$ is described by the automatic gain control [54]. The filter bandwidth $B$ is calculated by:

$$B = 1.019 \cdot 24.7 \cdot \left(\frac{4.37 f_0}{1000} + 1\right) \tag{4}$$

The center frequency values are spaced on the equivalent rectangular bandwidth scale [55,56]. The number of filters used in this research is equal to 50.

### 2.1.4. Fractal Dimension-Based Features

A fractal dimension-based analysis was employed for speech signal analysis considering the nonlinearity of emotional speech signals. Fractal dimension characterizes and differentiates the irregularity, self-similarity, and nonlinearity between different speech emotions. In this study, we selected fractal dimension-based features, whose effectiveness in classifying speech emotion was justified in a previous study by Katz, Castiglioni, and Higuchi on fractal dimensions [6].

### 2.2. CNN Architecture

As mentioned earlier, the convolutional neural network class of the DNNs was applied in this experiment, and as mentioned earlier, feature dimensionality has been selected according to its significance, and then resulting images were scaled to the same size, and the input for the classifier was fixed. The CNN network topology is a compound of three convolutional layers after each of those activation functions "relu", max pooling of the sizes 3 by 3 (the last layer uses max pooling size 2 by 2), and batch normalization was applied. Then, the flatten layer of the size 64 and batch normalization were added, and the "relu" layer activation function was utilized. The last layer acts as an emotion classifier with the "softmax" activation function. Adam optimization algorithm was applied to solve the optimization part of the emotion recognition experiment. The use of the batch normalization technique, together with the additive white noise (described in the next section), helps to prevent network overfitting. A more detailed description of the network is presented in the article published by the authors [49].

## 3. Datasets

The datasets were created by imitating real-world principles to some extent: different language data were collected without any assumption or pre-emphasis applied to the signal. Moreover, the data were not normalized or balanced, i.e., a different number of speakers, utterances, sentences were employed for several languages. In addition, all the data were augmented and processed automatically by employing unified algorithms. The primary assumption was that the data collection process should

be similar to the one that is common for real-world services. However, it should clearly be stated that all datasets consist of emotional speech containing recordings of non-professionally acting speakers.

For this study, six language emotion datasets were chosen: Lithuanian (Lithuanian Spoken Language Emotions Database recordings) [16], English dataset, which consists of recordings of three independent databases, i.e., RAVDESS (24 professional female and male actors, vocalizing two lexically-matched statements in a neutral North American accent) [17], SAVEE (this database consists of recordings from four male actors expressing seven different emotions) [18], and TESS (two English-native actresses) [19], Polish dataset that includes numbers and separate command word records of Polish Emotional Speech Dataset [24,25], Spanish (Spanish emotional speech synthesis database recordings) [22], Serbian (Serbian emotional speech GEES database recordings) [21], and German (Berlin emotional speech database recordings) [23].

These datasets share the same five archetypal primary emotions, which are anger, sadness, fear, neutral, happiness [48]. In this study, TESS, SAVEE, and RAVDESS datasets are merged in order to get one big English language dataset. A summarized statistic of the datasets is given in Table 2.

**Table 2.** Corpora information: sampling number for each emotion.

| Dataset | Emotions | | | | |
|---|---|---|---|---|---|
| | **Anger** | **Sadness** | **Fear** | **Neutral** | **Happiness** |
| Lithuanian | 1000 | 1000 | 1000 | 1000 | 1000 |
| English | 652 | 652 | 652 | 556 | 652 |
| Serbian | 180 | 180 | 180 | 180 | 180 |
| Spanish | 724 | 728 | 735 | 734 | 732 |
| German | 127 | 62 | 69 | 79 | 71 |
| Polish | 164 | 176 | 147 | 197 | 192 |

To augment the datasets (due to the CNN prerequisite to have a large amount of data to learn from), two following methods were used: the white noise of seven levels (0 dB, 5 dB, 10 dB, 15 dB, 20 dB, 25 dB, and 30 dB) was added; Wiener filtering was applied, thus providing cleaned recordings. This enabled us to increase data variability to decrease overfitting and expand our datasets (Table 2) as much as nine times.

## 4. Experimental Results

An experimental investigation consisted of two parts. To ensure that the process of the experiment can easily be followed, a block diagram of the experiment is shown in Figure 1.

As we see from Figure 1, in the first part of the experiment, various feature space-based emotion recognition was carried out on separate databases. The goal of this part was to check the quality of training for each of the datasets used in the experiment. All the datasets were split according to the rule 80/20 (80% of data were used for the model training/validation and 20% for testing). In addition, 80% of the dataset was split according to the same scheme, i.e., 80% were used in the model training and 20% for the model testing. Test sets contain samples of the same emotion uttered by the same speaker, with a repetitive emotion sample recording. To evaluate the classifier performance, both the overall network accuracy and the averaged F1 score were calculated. The results are given in Table 3.

Table 3 shows that cochleagrams enabled reaching the overall test accuracy (Test Acc.) and F1 score equal to 1 or 0.99, respectively. The spectrograms produced test output values around the average of 0.89, for mel-cepstrograms, it reached 0.87, and for fractal dimension-based features, it was 0.78. In addition, in Table 3 training accuracy (Training acc.) as well as validation accuracy (Validation Acc.) values are shown.

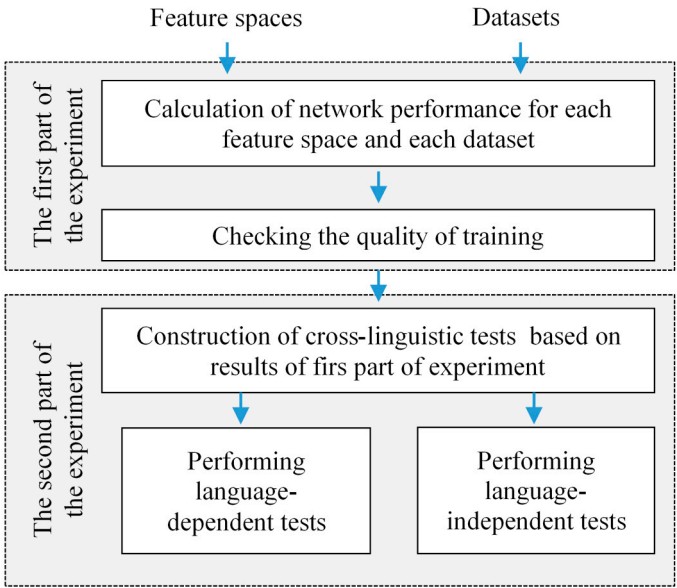

**Figure 1.** A block diagram of the experiment.

**Table 3.** Classification results.

| Dataset | Training Acc. | Validation Acc. | Test Acc. | F1 Score |
|---|---|---|---|---|
| *Spectrograms* | | | | |
| Lithuanian | 0.9636 | 0.8215 | 0.82 | 0.82 |
| English | 0.9999 | 0.9475 | 0.95 | 0.95 |
| Serbian | 0.9971 | 0.8981 | 0.91 | 0.91 |
| Spanish | 1 | 0.9471 | 0.94 | 0.94 |
| Polish | 1 | 0.8189 | 081 | 0.81 |
| German | 1 | 0.9282 | 0.92 | 0.91 |
| *Mel-cepstrograms* | | | | |
| Lithuanian | 0.8794 | 0.7453 | 0.76 | 0.76 |
| English | 0.9999 | 0.9659 | 0.97 | 0.97 |
| Serbian | 0.9969 | 0.9537 | 0.94 | 0.94 |
| Spanish | 0.9997 | 0.8984 | 0.91 | 0.90 |
| Polish | 0.9994 | 0.7768 | 0.77 | 0.76 |
| German | 1 | 0.9726 | 0.96 | 0.96 |
| *Fractal dimension-based features* | | | | |
| Lithuanian | 0.9469 | 0.6581 | 0.65 | 0.65 |
| English | 0.9252 | 0.8109 | 0.83 | 0.82 |
| Serbian | 0.8841 | 0.8093 | 0.85 | 0.83 |
| Spanish | 0.9301 | 0.7293 | 0.74 | 0.74 |
| Polish | 0.9363 | 0.7172 | 0.71 | 0.71 |
| German | 0.9702 | 0.8820 | 0.85 | 0.85 |
| *Cochleagrams* | | | | |
| Lithuanian | 0.9999 | 0.9681 | 1 | 1 |
| English | 1 | 0.9991 | 1 | 1 |
| Serbian | 0.9963 | 0.9938 | 1 | 1 |
| Spanish | 1 | 0.9971 | 1 | 1 |
| Polish | 1 | 0.9872 | 0.99 | 0.99 |
| German | 1 | 1 | 1 | 1 |

The results obtained from the first part of the experiments let us conclude that all datasets and all feature spaces can be used in the second part of the experiment. This part of the experiment corresponds

to testing the effectiveness of the signal feature spaces in a cross-linguistic speech emotion recognition task. To this end, training and validation datasets are the same for all testing datasets (as described earlier). Two types of tests were investigated: language-dependent and language-independent. The language-dependent tests were performed on 20% of data for each emotion database we had used in the network training. Test data were not utilized for network training and validation. For the language-independent test, we employed datasets that were not a part of the neural network training.

The classification results revealed the same tendency as in the first part of the experiments. The highest classification accuracies were achieved for the same feature space, i.e., cochleagrams. However, in terms of classification accuracy, a spectrogram-based representation also returns very high scores. The results of these two feature spaces are shown in Tables 4–7. The datasets that were used in the training process are highlighted in bold font.

**Table 4.** Classification test results for spectrograms and cochleagrams: Lithuanian dataset is used for training. The datasets that were used in the training process are highlighted in bold font.

|  | Spectrograms | | Cochleagrams | |
| --- | --- | --- | --- | --- |
|  | **Test Acc.** | **F1 Score** | **Test Acc.** | **F1 Score** |
| **Lithuanian** | **0.82** | **0.82** | **0.97** | **0.97** |
| English |  |  | 0.27 | 0.26 |
| Serbian | 0.37 | 0.18 | 0.41 | 0.36 |
| Spanish | 0.3 | 0.19 | 0.35 | 0.2 |
| Polish | 0.21 | 0.17 | 0.20 | 0.19 |
| German | 0.49 | 0.27 | 0.42 | 0.40 |

**Table 5.** Classification test results for spectrograms and cochleagrams: Lithuanian, English, and Serbian datasets are used for training. The datasets that were used in the training process are highlighted in bold font.

|  | Spectrograms | | Cochleagrams | |
| --- | --- | --- | --- | --- |
|  | **Test Acc.** | **F1 Score** | **Test Acc.** | **F1 Score** |
| **Lithuanian** | **0.85** | **0.85** | **0.95** | **0.95** |
| **English** | **0.94** | **0.94** | **0.99** | **0.99** |
| **Serbian** | **0.91** | **0.91** | **0.98** | **0.98** |
| Spanish | 0.35 | 0.34 | 0.31 | 0.30 |
| Polish | 0.33 | 0.21 | 0.30 | 0.25 |
| German | 0.55 | 0.52 | 0.51 | 0.49 |

**Table 6.** Classification test results for spectrograms and cochleagrams: Lithuanian, English, Serbian, and Spanish datasets are used for training. The datasets that were used in the training process are highlighted in bold font.

|  | Spectrograms | | Cochleagrams | |
| --- | --- | --- | --- | --- |
|  | **Test Acc.** | **F1 Score** | **Test Acc.** | **F1 Score** |
| **Lithuanian** | **0.82** | **0.82** | **0.88** | **0.88** |
| **English** | **0.92** | **0.92** | **0.94** | **0.94** |
| **Serbian** | **0.87** | **0.87** | **0.94** | **0.94** |
| **Spanish** | **0.9** | **0.9** | **0.94** | **0.94** |
| Polish | 0.34 | 0.34 | 0.26 | 0.24 |
| German | 0.56 | 0.49 | 0.58 | 0.53 |

**Table 7.** Classification test results for spectrograms and cochleagrams: all datasets are used for training. The datasets that were used in the training process are highlighted in bold font.

|  | Spectrograms | | Cochleagrams | |
| --- | --- | --- | --- | --- |
|  | **Test Acc.** | **F1 Score** | **Test Acc.** | **F1 Score** |
| **Lithuanian** | **0.83** | **0.83** | **0.89** | **0.89** |
| **English** | **0.92** | **0.92** | **0.96** | **0.96** |
| **Serbian** | **0.89** | **0.89** | **0.94** | **0.94** |
| **Spanish** | **0.91** | **0.91** | **0.95** | **0.95** |
| **Polish** | **0.76** | **0.76** | **0.88** | **0.88** |
| **German** | **0.88** | **0.88** | **0.95** | **0.94** |

For the classification results presented in Table 4, the Lithuanian dataset was used for training and language-dependent test. The network training accuracy is equal to 0.9636, and the validation accuracy is 0.8215 for spectrograms. In the cases of the cochleagrams, training and validation accuracies are 0.9999 and 0.9681, respectively. The remaining datasets were used for the language-independent test.

The results, when Lithuanian, English, and Serbian datasets were used for the network training, are shown in Table 5. In this case, the network performance is as follows: for spectrograms, training accuracy reaches 0.9628, validation accuracy is 0.8807, for cochleagrams, training accuracy is 0.9967, and the validation accuracy—0.9677.

The results shown in Table 6 include classification accuracies in the case of Lithuanian, English, Serbian, and Spain datasets used as training sets. For spectrograms, the training accuracy is 0.9462, validation accuracy equals 0.8641, for cochleagrams, the training accuracy is 0.9623, validation accuracy equals 0.9295.

The experiment with all datasets used for training was carried out as well. The results are given in Table 7.

Network training and validation accuracies are as follows (see Table 7): 0.9450 and 0.8623, respectively, for spectrograms, 0.9587, and 0.9265 for cochleagrams.

Mel-cepstrograms and fractal dimension-based features yielded similar classification tendencies. As the classification results of these two feature spaces are slightly worse than those of spectrograms, we decided not to recall them here.

There are additional studies on SER to compare our investigation performance with others. Our investigation is similar, to some extent, to the work of Fu et al. [39] and Neumann and Vu [40].

Fu et al. [39] used the same emotion labels (angry, fear, happy, neutral, sad) and trained the model using combinations of various corpora (German, Chinese, and French). For emotion recognition, authors used a neural network combining a one-dimensional CNN and bi-directional long short-term memory network (LSTM) network with an attention mechanism. Authors employed eleven different features concatenated in the one-dimensional 34th order feature vector in contrast to our investigated two-dimensional feature spaces, and most importantly, authors focus and depict results for different training methods. Thus, as the conditions are different, it is not possible to directly compare the results reported by Fu et al. [39] with those obtained in this study. The results of both studies can be characterized as scattered. This is inevitable for cross-lingual speech emotion recognition.

Neumann and Vu [40] present results for cross-lingual and multilingual emotion recognition as well. Authors employ English, French speech corpora, and the attentive convolutional neural network (ACNN), with the 26 logMel filter-bank output as and input features. The categorical emotion labelling is focused on a binary classification task of arousal (low/high) and valence (negative/positive) of the emotional speech. The facts lead to the same conclusion that it is not possible to directly compare the results presented by Neumann and Vu [40] to those obtained in our study.

We would emphasize the following moments of our study as advantages:

- Two-dimensional feature maps enable us to deliver temporal information in addition to the selected acoustic features of the emotion. This should save the emotion global (suprasegmental) features

for the analysis in the network and enhance the decision process. The sequential framing-based analysis of the speech prevents this, as all suprasegmental information is lost.

- We extract the same features for all the languages we analyze. This analysis scheme does not require any merging, mixing or any other joint processing of different language data. Furthermore, our feature maps are based on the single analysis technique, but we cannot deny the probable need to join different analysis techniques to improve the discriminative power of the feature maps.

- We have analyzed six different languages, which makes our study truly multilingual. The results of the cross-lingual emotion recognition are not impressive, but they reveal very clear challenges in the future: the need of multilingual emotional speech data (especially for low-resource languages), the undefined feature systems for the multilingual emotions, and the possible variation of the results for different languages.

## 5. Conclusions

With the same network architecture, the best training, validation, and test performance results were achieved by the cochleagram-based representation. This feature space enables us to reach the general test accuracy and F1 score on separate databases equal to 1 or 0.99 (in the case of Polish language). In a cross-linguistic speech emotion recognition task, test accuracy and F1 score vary from 0.88 to 0.96 in the case of all datasets (Lithuanian, English, Serbian, and Spanish, Polish, and German) as training sets.

It may be assumed that the best results in classification occurred for the cochleagram feature space because of one more dimension that it has, compared to other feature spaces. It may be that in this axis, which is orthogonal to the frequency axis, features associated with human emotions are contained. However, this statement requires a thorough analysis, which we should follow in future research. One additional suggestion in terms of the methodology would be to attempt to fuse a variety of the employed features as opposed to evaluating each feature space separately.

The experimental results depicted that emotions are language dependent. However, increasing the number of languages in a training set, the language-independent classification (for a language not used in the training step) accuracy slightly increases. In future research, the authors plan to expand the number of datasets for different languages so that it can be possible to state if it is possible to achieve high emotion recognition results for languages not used in the training process.

There is an additional conclusion related to this research study regarding data preparation. We have used the Polish database in tests, as well. However, the problem was that tagging was performed only on words due to the lack of automatic annotation of this database. This resulted in the lower classification accuracies obtained for this language compared to other languages. Therefore, we came to the obvious conclusion: very short utterances do not contain attributes of the spoken emotion. Thus, the next step should be to work on a strategy of the multilingual parser to acquire additional data from the speech utterances available on the Internet and prepare them automatically for parameterization and classification.

Finally, a question arises whether a ground truth database may be formulated containing emotionally "loaded" utterances, utilizing such techniques as, e.g., crowdsourcing [57] applied for both "producing" emotions in speech as well as evaluating gathered utterances. Such an experiment may result in more reliable datasets for the in-depth training process.

In future research, we intend to explore other factors, such as, e.g., speech phoneme properties (like duration, accent, energy), which may also affect the quality of recognition. We will investigate whether the considerable gap between emotional recognition for seen and unseen language exists because of different language properties or because of non-stationary and nonlinear speech signal properties. In addition, for data augmentation, the combination of STFT-based and Hilbert–Huang transform (HHT)-based features may be used as such an approach may improve the overall performance of a classifier [58]. Moreover, future research will investigate the problem of cross-language speech emotion recognition in terms of domain adaptation and transfer learning. Though, it should be mentioned

that data from different datasets were not reemphasized or balanced. That means that the task of transfer learning for emotion recognition is more difficult than it has to be because apart from language, there are also differences in recording conditions and dataset representation.

**Author Contributions:** Conceptualization, B.K. and G.K.; methodology, P.T. and G.T.; software, A.B.Y.; validation, G.K., J.B. and A.B.Y.; formal analysis, B.K.; investigation, G.T., P.T., G.K., J.B.; resources, P.T.; data curation, G.K., J.B.; writing—original draft preparation, G.K. and J.B.; writing—review and editing B.K.; visualization, A.B.Y.; supervision, P.T. All authors have read and agreed to the published version of the manuscript.

**Funding:** This research received no external funding.

**Acknowledgments:** The authors are thankful for the high-performance computing resources provided by the Information Technology Open Access Center at the Faculty of Mathematics and Informatics of Vilnius University Information Technology Research Center.

**Conflicts of Interest:** The authors declare no conflict of interest.

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
