# Peer review of "A Study of Cross-Linguistic Speech Emotion Recognition Based on 2D Feature Spaces"

_electronics, doi:10.3390/electronics9101725_

Round 1

Reviewer 1 Report

The manuscript is centered on an interesting topic. Organization of the paper is good and the proposed method is quite novel.

The manuscript, however, does not link well with recent literature on sentiment analysis appeared in relevant top-tier journals, e.g., the IEEE Intelligent Systems department on "Affective Computing and Sentiment Analysis". Also, latest trends in multilingual sentiment analysis are missing, e.g., see Lo et al.’s recent survey on multilingual sentiment analysis (from formal to informal and scarce resource languages). Finally, check recent resources for multilingual sentiment analysis, e.g., BabelSenticNet.

Some parts of the manuscript may result unclear for some Electronics readers. A short excursus on emotion categorization models and algorithms could resolve this lack of clarity (as the journal does not really feature many papers on this topic) and improve the overall readability of the paper.

On a related note, the manuscript presents some bad English constructions, grammar mistakes, and misuse of articles: a professional language editing service is strongly recommended (e.g., the ones offered by IEEE, Elsevier, and Springer) to sufficiently improve the paper's presentation quality for meeting Electronics’ high standards.

Author Response

Thank you for giving us the opportunity to improve our manuscript according to the Reviewers’ comments. We are very obliged to receiving these remarks. They allowed us to revise our paper.

We are sending an updated manuscript with red font color indicating changes, and a clean revised manuscript without highlights.

Response to the Reviewers:

  1. Review of recent literature on sentiment analysis was updated in Introduction section (table 1) and in Experimental Results section (lines: 281-312).
  2. A short review on emotion categorization models and algorithms is included in Introduction (lines: 39-58).
  3. English was improved (an updated manuscript with red font color indicating changes).

Once again, we wish to express our thanks for the opportunity to revise this manuscript.

With best regards,

Jolita Bernatavičienė and co-authors

Reviewer 2 Report

The research paper describes an emotion recognition method based on acoustic features. The study focused on cross-lingual emotion recognition because existing studies are primarily based on mono-lingual emotion recognition. Additionally, the research used two-dimensional acoustic features and CNN for training the dataset. The results obtained showed accuracy above 90% and cochleagrams features contributed more significantly when compared to other features. Overall the manuscript is well written with satisfactory explanation for motivation, literature study, method, results and conclusion. The future studies would include features based on accents, speech duration and energy. The study provides good insights into multi linguistic emotional expression and evaluated the proposed method on various languages such as English, German, Lithunian, Spansih, Polish and Serbian. As a result, the reviewer recommends a decision of accept.

Author Response

(The authors gave the same response as above.)
